# Multiplex PCR Pneumonia Panel in Critically Ill Patients Did Not Modify Mortality: A Cohort Study

**DOI:** 10.3390/antibiotics14030245

**Published:** 2025-02-28

**Authors:** Luisa Fernanda Riaño-Sánchez, Carlos Arturo Alvarez-Moreno, Marcela Godoy, Claudia Rocío Sierra, Margarita Inés Castañeda, Jorge Alberto Cortés

**Affiliations:** 1Departamento de Medicina Interna, Facultad de Medicina, Universidad Nacional de Colombia, Bogotá 111321, Colombia; lrianos@unal.edu.co (L.F.R.-S.); caalvarezmo@unal.edu.co (C.A.A.-M.); 2Clínica Reina Sofía, Clínica Colsanitas, Bogotá 110121, Colombia; 3Laboratorio Clínico y de Patología, Clínica Colsanitas, INPAC Research Group, Keralty Group, Bogotá 111131, Colombia; magodoy@colsanitas.com (M.G.);; 4Departamento de Terapias, Clínica Universitaria Colombia, Clínica Colsanitas, Bogotá 111321, Colombia; 5Hospital Universatario Nacional, Bogotá 111321, Colombia

**Keywords:** pneumonia, molecular diagnosis, multiplex polymerase chain reaction, intensive care unit, antimicrobial stewardship

## Abstract

In critically ill patients, identification of the pathogen may allow for the timely adjustment of antibiotics and improved outcomes. **Background/Objectives**: The aim of the study was to assess whether performing a multiplex PCR pneumonia panel (PN-panel) in patients with pneumonia in the intensive care unit (ICU) had any effect on mortality or other important clinical outcomes. **Methods**: A retrospective cohort study was conducted on adult patients with pneumonia who required ICU admission in four institutions in Bogotá between November 2019 and June 2023. Mortality at 30 days, the length of the hospital and ICU stay, the duration of antibiotics, and their association with the PN-panel performance were evaluated using an inverse probability of the treatment weighting to adjust for covariates and potential confounders. **Results**: A total of 304 patients were included, including 150 with PN-panel, with a mean age of 65.0 years (SD 14.6). SARS-CoV-2 was the primary etiologic agent in 186 (61.2%) patients, and 256 (84.2%) patients had community-acquired pneumonia. No association was found between 30-day mortality and the PN-panel, with a HR of 1.14 (CI 95% 0.76–1.70), although the assessment by an infectious disease specialist was associated with a lower mortality HR of 0.29 (CI 95% 0.19–0.45). There was no association between the PN-panel and antimicrobial therapy duration or other clinical outcomes. **Conclusions**: The use of the PN-panel was not associated with changes in mortality, the duration of antibiotics, or hospital and ICU stays. To acquire greater rational decision-making, microbiological data produced by this test should be interpreted with aid of an antimicrobial stewardship program oriented by an infectious disease team that could take the clinical data and integrate the information provided.

## 1. Introduction

Severe pneumonia is one of the main causes of hospitalization, affecting all age groups [1], and 10–23% of cases require attention in intensive care units (ICUs) [2,3], which has an impact on associated costs and mortality. Associated mortality can reach 26% in the general population [4], but it reaches 40% in adults over 80 years of age [5] and can reach 50% when a follow-up is performed up to one year after admission [6]. In turn, patients with severe pneumonia are at greater risk of infection by resistant microorganisms, particularly methicillin-resistant *Staphylococcus aureus* (MRSA) and *Pseudomonas aeruginosa*, among other resistant Gram-negative bacilli [7]. The diagnosis of pneumonia is clinical, and in most cases, microbiological identification of the causative agent is not achieved with conventional methods [8]. Traditionally, identification does not impact relevant outcomes; therefore, antibiotics are directed according to the recommendations of management guidelines [9,10]. Previous guidelines have recommended molecular identification, specifically for suspected Influenza infections [11], but more recent guidelines proposed by multiple scientific associations have recommended adding molecular tests, when available, for patients with severe pneumonia [11] because of the potential relevance for timely adjustments in antimicrobial coverage [12]. Given that it has been shown that in the community sepsis scenario, inappropriate and unnecessary broad-spectrum coverage impacts mortality [13], the use of tools such as multiplex Polymerase Chain Reaction (PCR) syndromic panels has recently increased [14]. There is evidence of good diagnostic performance of these tests, with positive agreement of approximately 90% and negative agreement of close to 100%, as well as superior sensitivity to cultures [15]. However, no consistent evidence indicates that its use modifies the outcomes of interest to the patient, such as the hospital stay, intensive care unit stay, and mortality. In addition, concerning the antibiotic changes, there are controversial data with variable adjustment potentials [16].

The objective of this study was to determine whether the use of a multiplex PCR panel for pneumonia (PN-panel) (Filmarray^®^, Biomérieux, Marcy-l’Étoile, France) is related to significant outcomes, such as the hospital stay, intensive care unit stay, duration of antimicrobial treatment, and 30-day mortality in patients with severe pneumonia in the ICU through a retrospective cohort between 2019 and 2023. A secondary outcome was to evaluate the effects of the infectious disease assessment, for which adjustments were evaluated according to PN-panel reports and/or cultures.

## 2. Results

A total of 842 eligible patients were identified, 534 were excluded, 304 patients were evaluated, 150 were exposed, and 154 were unexposed (Figure 1). The mean age was 65 years old (SD 14.6), the mean Charlson comorbidity index was 4.5 (SD 2.8), and the most frequent comorbidities are presented in Table 1.

### 2.1. Characteristics of Pneumonia and Admission to the ICU

The most frequent type of pneumonia was community-acquired in 257 (84.5%) patients, of whom 122 (81.3%) used the PN-Panel and 134 (87%) did not. Hospital-acquired pneumonia (HAP) accounted for 40 cases (13.1%), and 8 (2.6%) were events of ventilation-associated pneumonia (VAP). COVID-19 was diagnosed in 186 patients (61.2%) (Table 1).

After inverse Probability Treatment Weighting (IPTW), the majority of variables were balanced, but the following relevant variables were not balanced: the Charlson score, history of chronic obstructive pulmonary disease (COPD) or chronic kidney disease (CKD), and APACHE-II score (Table 1 and Appendix A).

### 2.2. Microbiological Isolates

Respiratory cultures were performed in 153 patients (50.3%), with 136 (44.7%) samples of tracheal aspirate, 9 (3%) of bronchoalveolar lavage (BAL), and 8 (2.6%) of sputum. The most frequently isolated microorganisms in respiratory cultures were *Klebsiella pneumoniae* and *Staphylococcus aureus*, 32 (20.9%) and 21 (13.7%), respectively. (See Appendix A for the complete list). Blood cultures were taken from 211 patients (69.4%). However, of the total blood cultures taken, 56 (26.5%) were positive. The most frequent isolate was coagulase-negative staphylococci, especially *Staphylococcus epidermidis* (13, 6%), which were usually interpreted as contaminations, followed by *K. pneumoniae* (10, 4.6%) (Appendix A). For those exposed, the most frequent amplifications in the PN-panel were *S. aureus* in 39 samples (26%), *K. pneumoniae* in 29 (19.3%), and *Escherichia coli* in 26 (17.3%). The resistance genes that were most frequently amplified corresponded to mecA/C + staphylococcal cassette chromosome mec (SCCmec)-orfX right-extremity junction (MREJ) in 16 samples (16.7%), KPC in 11 (7.3%), and CTX-M in 10 (6.7%). Among the viral targets, the most frequent was Rhinovirus/enterovirus in 11 samples (7.3%) (see Appendix A for the complete list).

### 2.3. Assessment by Infectious Diseases (ID) Team and Use of Antibiotics

In total, 222 patients were seen by the ID team (73.0%), and patients with the PN-panel were seen more frequently by this team (82.0% vs. 64.3%). There were other differences among patients that were seen by the ID team: patients were younger (63.5 vs. 69.2 years old, *p* = 0.003), had, more frequently, ground glass in the chest X-ray (56.8% vs. 41.5%, *p* = 0.025), had, more frequently, a blood culture taken (76.1% vs. 51.2%, *p* < 0.001), and had a lower frequency of community-acquired pneumonia (81.1% vs. 92.7%, *p* = 0.033). Patients with COVID-19 were less frequently seen by the ID team (68.8% vs. 79.6%, *p* = 0.052).

Changes in the antimicrobial management by the ID team were performed more frequently in the group of the PN-panel than in the control group (90 cases, 60%, vs. 49 cases, 31.8%, *p* < 0.001). Additionally, the evaluation by the ID team allowed us to scale the therapy at 20.3% vs. 2.4% (*p* < 0.001) and suspend the therapy at 13.5% vs. 1.2% (*p* = 0.003) (Table 2).

### 2.4. Primary Outcomes

Antibiotic use, changes and adjustments, and the outcomes observed in the overall cohort are shown in Table 3.

After adjustment with IPTW, the great majority of variables achieved a balance (Table 1). Important differences in survival were noted among those patients who were assessed by the ID team (Figure 2). No association was found between mortality and PN-panel use, with an HR of 1.14 (95% CI 0.76–1.70). The ID team assessment was associated with lower mortality at 30 days, with an HR of 0.29 (95% CI 0.19–0.45) (Table 4).

The sensitivity analysis performed showed similar results when selecting only patients with CAP. Mortality risk at 30 days was similar among patients with a PN-panel HR of 1.00 (95% CI 0.64–1.56), and the effect of the ID intervention showed the same favorable trend, HR 0.27 (95% CI 0.17–0.43).

No association was found between PN-panel use and the duration of antibiotics (BN model), with an IRRa of 1.17 (95% CI 0.9–1.54) (Table 4). The hospital stays of the 145 patients who survived had a median of 18 days (IQR 12–34) overall; it was 22 days (IQR 12.5–37.5) for the 67 exposed patients and 16 days (IQR 10.3–32.5) for the 78 unexposed patients. No association was found between the PN-panel and hospital stay (NB model), with an IRRa of 1.02 (95% CI 0.75–1.38), nor was there an association with the duration of an antimicrobial IRRa of 1.04 (95% CI 1.02–1.05), whereas with the ID team assessment, an association was found with the increase in a hospital stay IRRa of 1.54 (95% CI 1.24–1.91) (Table 4).

The median ICU stay (145 patients who survived) was 10 days (IQR 6–17.5) for 67 exposed patients and 8 days (IQR 5–15) for 78 unexposed patients. No association was found between PN-panel use and ICU stays (NB model), with an IRRa of 1.24 (95% CI 0.8–1.9), or for the duration of antimicrobial use (NB model), with an IRRa of 1.03 (95% CI 1.01–1.05). The ID team assessment was related to an increase in ICU stays (NB model) (IRRa 1.35, 95% CI 1.03–1.76) (Table 4).

### 2.5. Secondary Outcomes

No association was found between changes in antibiotic use and PN-panel use, ORa (0.53; 95% CI 0.26–1.07), but the ID team evaluation was associated with a decrease in the number of antibiotics, with an ORa of 0.06 (95% CI 0.03–0.15).

The calculation of the percentages of agreements and kappa indices between the PN-panel and the respiratory sample cultures discriminated by microorganisms is shown in Appendix A. A positive agreement of 74.6% and an overall negative agreement of 95.2% were observed, with a kappa coefficient of 0.51. An agreement was assessed between the CFU and the number of copies/mL; the best percentage of agreement was found for negative reports, with 99.2% agreement, and for a CFU of 10^5^, with 76.2% agreement (Appendix A).

## 3. Discussion

This retrospective cohort of 304 patients with severe pneumonia who required intensive care unit (ICU) management between November 2019 and June 2023 were evaluated regarding the effectiveness of the BioMérieux Filmarray^®^ (Marcy-l’Étoile, France) multiple PCR panel for pneumonia over mortality and other main outcomes and found no associations between the use of this diagnostic test and the outcomes analyzed. Its use did not translate into a shorter hospital stay, a decrease in the use of antibiotics, or a lower mortality.

The absence of a significant impact of PN-panel use on 30-day mortality and hospital stays is consistent with the results of a randomized, multicenter, parallel, and superior study in Denmark that did not find a significant association between the intervention (PN-panel use) and mortality. At 30 days, the OR was 1.24 (95% CI 0.32–4.82), and the hospital stay IRR was 0.82 (95% CI 0.63–1.07) [16,18]. The absence of changes in the hospital stay was also reported in the results of the “Flagship II” clinical trial, in which a hospital stay of 14.4 days (SD 10.4) was identified in the PCR group vs. 16.0 days (SD 11.2) in the control group (*p* = 0.29) [19].

On the other hand, an ID team assessment reduced mortality and changes in antimicrobial therapy that were not based on the microbiological results (culture or molecular panel). The patients evaluated by the ID team had more adjustments, including discontinuation or the use of antimicrobials with a narrow spectrum. Despite the above, a higher frequency of reinitiation of antimicrobial therapies by an attending medical team was found without considering the recommendations given by the ID team or the microbiological results. This poor adhesion may reflect the behavior observed in countries such as Colombia, where the use of antibiotics is carried out in the context of fear of the patient’s outcome [20], and multiple restarts of antimicrobials are made by the attending physician during the same infectious event, despite the concepts of discontinuation or adjustment with defined times by the ID team.

There is controversy about the usefulness of molecular panels, particularly with exposure to unnecessary antibiotics or the duration of antimicrobial therapy. Data from observational studies have been reported in the literature, in which the spectrum is reduced, global antimicrobial exposure is decreased [16,21,22], the consumption of antibiotics is directed against resistant microorganisms [23] and, even with data from the “Flagship II” study, the duration of inappropriate therapies is reduced by 45% (95% CI 37.9–52.1) [19]. However, there are observational studies in which no or minimal changes occurred [24,25,26]. In this cohort, no association was found between PN-panel use and a reduction in the duration of antimicrobial therapy, similar to the results of a randomized clinical trial in France [27]. The usefulness of the panel in decision-making was evaluated in conjunction with the performance of procalcitonin as a point of care in the emergency room, but it failed to demonstrate that the performance of the panel reduced antimicrobial exposure [27].

Given the important findings on the role of the ID team, it is essential to enhance their role in multidisciplinary decision-making in patients with severe pneumonia. In addition, it is important to strengthen the multidisciplinary teams of antimicrobial stewardship programs (AMS programs) to improve reasoned decision-making according to microbiological results and promote the education of non-infectious clinicians in the rational interpretation of PN-panel reports, as data suggest that intensivists perceive the results of a multiplex PCR panel to be more credible for decision-making [28].

The ERS/ESICM/ESCMID/ALAT guide for the management of CAP published in 2023 left a recommendation to consider taking the test in scenarios where it is considered prudent to initiate empirical schemes other than those recommended by the management guidelines [11]. Based on the evidence and findings of the study, this recommendation could be adopted as part of a structured multidisciplinary AMS program.

On the other hand, the percentages of positive and negative agreements of the PN-panel with the cultures of respiratory samples were 74.6% and 95.2%, respectively, which are similar to the data reported by other studies [29,30,31]. The differences in agreement among microorganisms are also similar to those reported in previous studies, such as Buchan et al. [32], and contribute data to the diagnostic test’s performance. As it is a more sensitive diagnostic tool concerning respiratory cultures, the kappa index may be low and should be interpreted with care.

This study has limitations. It was not possible to match according to the criteria set by the characteristics of the patients and the participating institutions, which could increase the risk of selection bias. The sample size may be too small to assess minimal effects on some outcomes. Patients with COVID-19 were included, in whom an increase in the prescription of antimicrobials was observed [33], along with increased isolation and transmission of Gram-negative bacilli (GNB) with various types of resistance [34,35]. Consequently, the cohort had a predominance of *K. pneumoniae* and *S. aureus* isolates, which are associated with increased mortality [8]. Additionally, pneumonia is diagnosed under real-life conditions in clinical practice, so it is possible that erroneous diagnoses cannot be controlled in retrospective clinical studies of pneumonia and that, consequently, a proportion of patients have a differential diagnosis [36], which makes it more difficult to find meaningful results in small samples. Another limitation is that the samples for culture and the PN-panel might have been taken with a difference of up to 72 h. That might have had an impact on the culture results, especially in patients who received antibiotics. Notably, none of the samples with a *Streptococcus* species were identified using a culture method. That suggests a potential advantage of the molecular test for microorganisms with special nutritional requirements in the culture or for those that are difficult to grow.

Multiple questions remain to be answered; the population that can benefit the most from diagnostic tests based on multiplex PCR is not clear; therefore, it is necessary to continue studying the behavior of patients in ICUs with severe pneumonia with real-life studies and strict inclusion criteria that allow significant differences to be identified. In this regard, a randomized clinical trial was recently published on patients with CAP who required hospitalization but did not find any usefulness of the use of multiple PCRs in mortality or antimicrobial consumption [37] as another clinical trial of superiority in diagnostic optimization that showed no differences in hospital stays or mortality [38].

Finally, it is striking that meta-analyses in the literature have evaluated the usefulness of upper respiratory tract panels in reducing antibiotic therapy; one of them has shown no significant impact [39], and the other has shown an impact, particularly when the viral etiology [40] is evaluated, but not specifically in the context of pneumonia. With the available evidence regarding the impact of antibiotics on the AMS program, to reduce inappropriate therapies and antibiotic days, a meta-analysis of the existing information is needed to strengthen the tools of the AMS program and the ratio of the use of antimicrobials in severe pneumonia patients.

## 4. Materials and Methods

### 4.1. Study Data and Subjects

A retrospective cohort of adult patients older than 18 years with suspected pneumonia who were admitted to the ICUs of 4 high-complexity hospitals in Bogotá, Colombia, between November 2019 and June 2023 was established. Patients with suspected CAP, HAP, and ventilation-associated pneumonia (VAP) were included [41,42]. Diagnosis of pneumonia followed standard definitions, with the presence of respiratory symptoms, clinical signs of pulmonary compromise, and a chest X-ray or other lung image showing consolidation without a non-infectious diagnosis. Patients without appropriate information in the medical record to define the variables of interest were excluded, as well as patients referred during hospitalization to another nonparticipating hospital center and those in which the PN-panel was taken more than 72 h after the start of IMV. Once the eligible patients were identified, a retrospective review of medical records was performed, and the variables of interest were filled out in a REDCap clinical collection format (version 6.16.8, Vanderbilt University, Nashville, TN, USA).

### 4.2. Exposure, Microbiological Procedures, and Assessment by the ID Team

Exposure was defined as the analysis of a respiratory sample using the PN-panel in patients with pneumonia.

#### 4.2.1. Microbiological Procedures

Respiratory specimens of expectorated or induced sputum, tracheal secretions, or bronchoalveolar lavage fluid were processed for culture, detection, and identification of multiple viral and bacterial nucleic acids using the FilmArray^®^ Pneumonia Panel (Filmarray^®^, Biomérieux, Marcy-l’Étoile, France) (PN-panel). Processing and interpretation followed the manufacturer’s instructions [43]. Briefly, the PN-panel provides a Detected or Not Detected result. Bacteria are reported semiquantitatively, with ranges representing approximately 10^4^,10^5^, 10^6^, or ≥10^7^ genomic copies of bacterial nucleic acid per milliliter of sample.

Respiratory specimens were incubated according to standard microbiologic laboratory guidelines [44]. Quantities of bacterial growth above a threshold were considered diagnostic of pneumonia: (Endotracheal aspirates, 10^6^ CFU/mL; BAL, 10^4^ CFU/mL; protected specimen brush samples (PSB), 10^3^ CFU/mL. Standard microbiological procedures were applied to identify the isolates performed with MALDI-TOF MS (VITEK-MS, bioMerieux, Marcy-l’Étoile, France). The automated system (VITEK-XL, bioMerieux, Marcy-l’Étoile, France) was used to determine antibiotic susceptibility, and the Clinical Laboratory Standards Institute was followed for the interpretation of susceptibilities (CLSI) [45].

#### 4.2.2. ID Team Assessment

The ID team evaluated in two ways: first, by request by the treating group according to clinical criteria, and second, owing to the existence of antimicrobial optimization programs implemented in the institutions, in which the evaluation of broad-spectrum antimicrobial agents was performed, with an assessment of their indication and appropriateness. Protocols for pneumonia and COVID-19 diagnosis and management were similar in all the centers.

### 4.3. Outcomes

The primary outcome was the 30-day mortality calculated from the time of a diagnosis of pneumonia. Hospital stays and ICU stays were defined as the length of hospital stay and ICU stay, respectively, for surviving patients only. Antimicrobial duration was defined as the number of days from the initiation to completion of antimicrobial therapy for pneumonia.

Secondary outcomes included the effects of the infectious disease assessment, for which adjustments were evaluated according to PN-panel reports and/or cultures and were classified according to definitions established by the researchers (Appendix A). They also included the effect of PN-panel use on antibiotics changes, and finally, the concordance between the reports of amplified targets in the PN-panel and the isolates in respiratory cultures, for which it was guaranteed that the samples were taken simultaneously or with a maximum difference of 72 h.

### 4.4. Covariates and Definitions

Variables were identified as possible confounders since they could have a determining effect on PN-panel use or outcomes. These covariates included age, days of hospital stay prior to pneumonia diagnosis, asthma, diabetes mellitus (DM), dementia, heart failure, hypertension (HT), stroke, peripheral vascular disease, chronic obstructive pulmonary disease (COPD), autoimmune disease, peptic acid disease, chronic liver disease, obesity, myocardial infarction, solid organ malignancy, lymphoma, leukemia, multiple myeloma, interstitial lung disease (ILD), age-adjusted Charlson comorbidity score, shock on ICU admission, respiratory failure on ICU admission, alveolar opacities, interstitial lung abnormalities, pleural effusion as radiological findings upon admission to the ICU, sepsis, SOFA score, APACHE-II score upon admission to the ICU, CAP, bronchoaspiration, use of steroids, COVID-19, respiratory distress syndrome (ARDS), pulmonary embolism (PE),use of mask with reservoir (MR) at ICU admission, MR as maximum support in the ICU, nasal cannula at ICU admission, nasal cannula as maximum ICU support, high-flow nasal cannula (HFNC) at ICU admission, HFNC as maximum ICU support, mechanical ventilation (MV) at ICU admission, MV as maximum support in the ICU, septic shock and leukocytes, lymphocytes, oxygen saturation (O2 Sat), and C-reactive protein (CRP) on admission to the ICU.

Appropriate antibiotic use was defined according to a microbiological isolate if available and susceptibility testing, the absence of a mechanism of resistance or a specific recommendation for an identified mechanism of resistance in those that only had PN-panel identification, and the recommended antibiotic according to the guideline if no further microbiological information was available.

An empiric antibiotic was defined as the use of an antibiotic based on the clinical diagnosis without identification of the causative agent of the infectious episode. A narrow-spectrum antibiotic was defined as the use of antibiotics that have an effect on a lower number of microorganisms or are intended to affect a specific genus or species.

### 4.5. Statistical Analysis

Continuous variables were described as the means and standard deviations (SDs) for variables with normal distribution, or median and interquartile ranges (IQRs) for the other variables. For categorical variables, data were presented as percentages and frequencies. In order to analyze the outcomes, inverse probability treatment weighting (IPTW) was used to control biases. Initially, logistic regression was performed with the 49 covariates to execute a propensity score and assign stabilized weights that would allow balancing according to the inverse probability of having used the PN-panel. The standardized mean differences (SMDs) of the original population and the resulting pseudopopulation were calculated to verify the balance [46]. The visual evaluation was carried out via a stabilized weight graph, box-and-whisker graphs, and a dot graph with the SMDs. For the 30-day mortality analysis, a Cox proportional hazards model adjusted by the IPTW was used, and the PN-panel use and other variables that could have a potential confounding effect were included, given that they had not been used in the IPTW or had not reached the necessary adjustment. HR was calculated with confidence intervals with robust errors. A sensitivity analysis was performed for the mortality results, taking into account only patients with CAP (256 patients, 84.2%). For this group, the same IPTW procedure was followed. For the analysis of hospital stays, ICU stays, and the duration of antimicrobial therapy, count models (NB) adjusted by IPTW were used, with the models with the lowest dispersion and the lowest error (AIC) being chosen. PN-panel use, infectious disease assessments, and antimicrobial durations were included in the hospitalization models, and the IRRa values were calculated with confidence intervals with robust errors. To determine the effect of infectious disease assessments and the relationship between antibiotic changes and PN-panel use, a multivariate logistic regression model was performed via IPTW. Finally, for the assessment of agreement between the PN-panel reports and respiratory cultures, percentages of agreement and Cohen’s kappa index were calculated. The sum of samples was taken as the denominator for the percentages of agreement. All statistical analysis was carried out using R (version 4.0.6, Vienna, Austria). Statistical significance was defined as a *p*-value < 0.05.

## 5. Conclusions

The use of a PN-panel was not associated with a reduction in mortality at 30 days, neither in hospital stays nor in the ICU. It also did not reduce the duration of antimicrobial therapy. The infectious disease assessment is related to a reduction in mortality, a decrease in changes in nontargeted antibiotics, and significantly favored adjustments to scale up the therapy or to suspend it. The application of a PN-panel must be accompanied by clinical judgment and commitment to interpretation. These data reinforce the importance of the role of ID teams in the targeting of antimicrobial therapy and its potential utility in improving antimicrobial optimization programs.

## Figures and Tables

**Figure 1 antibiotics-14-00245-f001:**
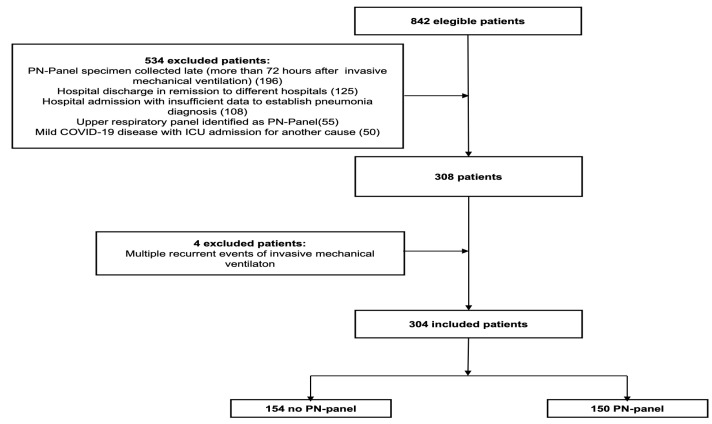
Flow diagram of patients included in the cohort adapted from [17].

**Figure 2 antibiotics-14-00245-f002:**
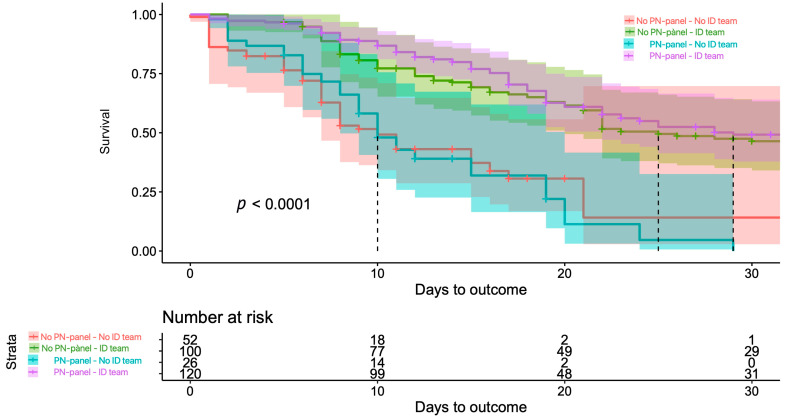
IPTW-adjusted 30-day mortality showing the effect of the use of PN-panel and ID team assessment adapted from [17].

**Table 1 antibiotics-14-00245-t001:** Characteristics of Individuals on Date of Diagnosis or PN-panel sample adapted from [17].

	Original Population				Post IPTW Pseudopopulation		
Variables	Total (*n* = 304)	PN-Panel (*n* = 150)	No PN-Panel (*n* = 154)	SMD	PN-Panel	No PN-Panel	SMD
Age (years), mean (SD)	65.0 (14.64)	63.6 (15.2)	66.4 (14.0)	0.196	65.0 (14.1)	64.64 (13.9)	0.03
Male, *n* (%)	206 (67.8)	105 (70.0)	101 (65.6)	0.095	98.3 (67.4)	98.8 (64.9)	0.052
**Comorbidity**							
Charlson comorbidity index, mean(SD)	4.5 (2.77)	4.7 (2.87)	4.3 (2.65)	0.164	4.6 (2.8)	4.2 (2.7)	0.14
Diabetes mellitus *n* (%)	103 (33.9)	50 (33.3)	53 (34.4)	0.023	48 (33.2)	47 (31.1)	0.043
Obesity, *n* (%)	82 (27.0)	34 (22.7)	48 (31.2)	0.193	35 (23.6)	38 (25.0)	0.032
Chronic kidney disease, *n* (%)	62 (20.4)	37 (24.7)	25 (16.2)	0.21	37 (25.5)	25 (16.4)	0.225
COPD *, *n* (%)	62 (20.4)	32 (21.3)	30 (19.5)	0.046	34 (23.3)	28 (18.3)	0.123
Heart failure, *n* (%)	58 (19.1)	31 (20.7)	27 (17.5)	0.08	25 (17)	24 (16.1)	0.026
Myocardial Infarction, *n* (%)	41 (13.5)	22 (14.7)	19 (12.3)	0.068	18 (12.3)	16 (10.3)	0.061
Solid organ malignancy, *n* (%)	38 (12.5)	22 (14.7)	16 (10.4)	0.129	23 (16.0)	21 (13.6)	0.07
**Characteristics of pneumonia**							
COVID-19, *n* (%)	186 (61.2)	68 (45.3)	118 (76.6)	0.677	87 (59.6)	96 (63.3)	0.082
Bronchoaspiration, *n* (%)	23 (7.6)	13 (8.7)	10 (6.5)	0.082	16 (10.8)	14 (9.2)	0.055
CAP *, *n* (%)	256 (84.2)	122 (81.3)	134 (87.0)	0.196	121 (82.7)	130 (85.5)	0.078
**Characteristics of ICU admission**							
SOFA, mean (SD)	6.5 (3.9)	7.43 (3.9)	5.62 (3.6)	0.482	6.7 (3.88)	6.7 (3.92)	0.012
Sepsis, *n* (%)	180 (59.2)	93 (62.0)	87 (56.5)	0.112	89 (61.3)	93 (61.4)	0.003
Shock, *n* (%)	141 (46.4)	79 (52.7)	62 (40.3)	0.251	69 (47.3)	69 (45.6)	0.034
Alveolar opacities, *n* (%)	174 (57.2)	89 (59.3)	85 (55.2)	0.084	89 (60.7)	92 (60.6)	0.003
Interstitial lung abnormalities, *n* (%)	143 (47.0)	71 (47.3)	72 (46.8)	0.012	71 (48.9)	73 (47.7)	0.025
HFNC *, *n* (%)	39 (12.8)	12 (8.0)	27 (17.5)	0.289	16 (10.8)	19 (12.7)	0.06
MV (IMV + NIMV) *, *n* (%)	105 (34.5)	63 (42.0)	42 (27.3)	0.313	55 (37.4)	52 (34.2)	0.067
**Features of ICU stay**							
Steroids, *n* (%)	241 (79.3)	117 (78.0)	124 (80.5)	0.062	115 (78.9)	126 (82.7)	0.096
ARDS *, *n* (%)	141 (46.4)	70 (46.7)	71 (46.1)	0.011	68 (46.7)	69 (45.1)	0.032
Lymphocytes (cells/μL), median (IQR)	815 (510–1175)	835 (518.5–1127.5)	800 (502.5–1220)	0.131	785 (498.1–1067.7)	760 (460–1170.4)	0.092
PaO_2_ * (mmHg), mean (SD)	66.8 (14.9)	68 (15.9)	65.6 (13.9)	0.161	67.42 (16.15)	67.0 (13.68)	0.028
C reactive protein (mg/L), mean (SD)	143.9 (113.9)	148.5 (122.3)	139.4 (105.4)	0.08	155.92 (112.35)	156.32 (116.54)	0.003

Continuous variables are represented as mean (standard deviation) and categorical variables as *n* (%). * COPD = Chronic Obstructive Pulmonary Disease, CAP = Community-acquired pneumonia, HFNC = High Flow Nasal Cannula, MV = Mechanical Ventilation, IMV = Invasive Mechanical Ventilation, NIMV = Non-Invasive Mechanical Ventilation, ARDS = Acute Respiratory Distress Syndrome, PaO_2_ = Oxygen arterial pressure.

**Table 2 antibiotics-14-00245-t002:** Results of ID team assessment.

	ID Team Assessment		*p*-Value
	Yes *n* = 222	No *n* = 82	
Adjustment with PN-panel results, *n* (%)	53 (23.9)	7 (8.5)	0.005
Adjustment with cultures results, *n* (%)	56 (25.2)	6 (7.3)	**0.001**
Decrease antimicrobial spectrum, *n* (%)	17 (7.7)	2 (2.4)	0.161
Increase antimicrobial spectrum, *n* (%)	45 (20.3)	2 (2.4)	**<0.001**
Discontinuation, *n* (%)	30 (13.5)	1 (1.2)	**0.003**
Antibiotic changes, *n* (%)	81 (36.5)	74 (90.2)	**<0.001**
Restart of antibiotics, *n* (%)	112 (50.5)	15 (18.3)	**<0.001**
Time to adjustment of antibiotics (days), mean (SD)	4.3 (4.8)	0.8 (1.3)	**0.003**
Number of antibiotics, mean (SD)	3.1 (1.7)	1.7 (1.3)	**<0.001**
Length of antibiotic therapy (days), mean (SD)	12.8 (8.7)	5.8 (7.1)	**<0.001**

Effect of ID team assessment in different activities related to antibiotic use. Statistically significant differents are bolded. Adapted from [17].

**Table 3 antibiotics-14-00245-t003:** Unadjusted primary and secondary outcomes.

	Total (*n* = 304)	PN-Panel (*n* = 150)	No PN-Panel (*n* = 154)	*p*-Test
Empiric antibiotics, *n* (%)	272 (89.4)	148 (98.7)	124 (80.5)	**<0.001**
Narrow spectrum antibiotics, *n* (%)	2 (0.7)	1 (0.7)	1 (0.8)	1
Appropriate antibiotic, *n* (%)	113 (75.3)	77 (52)	36 (28.8)	0.97
Adjustment with PN-panel results, *n* (%)	60 (19.7)	60 (40.0)	0 (0)	**<0.001**
Adjustment with cultures results, *n* (%)	62 (20.4)	37 (24.7)	25 (16.2)	**0.093**
ID team assessment, *n* (%)	222 (73.0)	123 (82.0)	99 (64.3)	**0.001**
ID adjustment of antibiotics, *n* (%)	130 (42.8)	80 (53.3)	50 (32.5)	**<0.001**
**Decision**				
Increase spectrum, *n* (%)	47 (15.5)	35 (23.3)	12 (7.8)	**<0.001**
Decrease spectrum, *n* (%)	19 (6.2)	14 (9.3)	5 (3.2)	0.051
Discontinuation, *n* (%)	31 (10.2)	15 (10.0)	16 (10.4)	1
Antibiotic changes, *n* (%)	155 (51.0)	51 (34.0)	104 (67.5)	**<0.001**
Restart of antibiotics, *n* (%)	127 (41.8)	79 (52.7)	48 (31.2)	**<0.001**
Number of antibiotics (mean-SD)	2.7 (1.7)	3.3 (1.7)	2.2 (1.6)	**<0.001**
Time to adjustment (days), mean (SD)	3.9 (4.71)	3.7 (4.2)	4.2 (5.3)	0.522
Length of antibiotic therapy (days), mean (SD)	10.9 (8.9)	12.5 (8.5)	9.4 (9.0)	**0.002**
Length of hospital stay (days), mean (SD)	20.8 (17.0)	22.7 (19.1)	18.9 (14.5)	0.056
Length of ICU stay (days), mean (SD)	12.2 (10.9)	13.1 (11.7)	11.3 (10.0)	0.153
Time to discharge (days), mean (SD)	18.7 (26.2)	20.1 (34.3)	17.5 (14.4)	0.385
In-hospital mortality, *n* (%)	159 (52.3)	83 (55.3)	76 (49.4)	0.353
30-day mortality, *n* (%)	147 (48.4)	77 (51.3)	70 (45.5)	0.362

Continuous variables are represented as mean (standard deviation) and categorical variables as *n* (%). ICU: Intensive care unit. Statistically significant data has been bolded.

**Table 4 antibiotics-14-00245-t004:** Adjusted primary and secondary outcomes.

Outcomes	Variables	Model	Estimator	95% CI	*p*-Value
**Mortality**			**HR**		
	PN-panel use	Cox proportional risks model	1.14	0.76–1.70	0.72
	ID assessment performed		0.29	0.19–0.45	<0.001
**Length of hospital stay**			**IRRa**		
	PN-panel use	Negative binomial (NB) model	1.02	0.75–1.38	0.83
	Length of antibiotic therapy (per day)		1.04	1.02–1.05	<0.001
	ID assessment performed		1.54	1.24–1.91	0.002
**Length of ICU stay**			**IRRa**		
	PN-panel use	NB model	1.24	0.8–1.9	0.134
	Length of antibiotic therapy (per day)		1.03	1.01–1.05	<0.001
	ID assessment performed		1.35	0.91–1.98	0.13
**Length of antibiotic therapy**			**IRRa**		
	PN-panel use	NB model	1.17	0.9–1.54	0.10
**Antibiotic changes**			**ORa**		
	PN-panel use	NB model	0.53	0.26–1.07	0.55
	ID team assessment performed		0.06	0.03–0.15	<0.001

95% CI: 95% Confidence interval; HR: Hazard ratio; IRRa: Adjusted incidence rate ratio; ORa: Adjusted odds ratio adapted from [17].

## Data Availability

Anonymized data are available on request.

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
