# Peer review of "Multiplex PCR Pneumonia Panel in Critically Ill Patients Did Not Modify Mortality: A Cohort Study"

_antibiotics, 2025, doi:10.3390/antibiotics14030245_

Round 1

Reviewer 1 Report

Comments and Suggestions for Authors

General concept comments

The topic addressed is both interesting and relevant, reflecting an important area of research. However, it is important to note that there are several critical aspects that require attention.

The authors provide an introduction that includes current references, but pneumonia or severe pneumonia clinical guidelines are not mentioned. Additionally, the use of abbreviations should be reviewed, as some are not defined in the text and they use some non-standard abbreviations, which may hinder the manuscript understanding.

There are significant concerns with the methodology employed that must be carefully addressed:

-       The authors used different timepoints to calculate the 30-day mortality (which is one of the primary outcomes); the pneumonia panel group used the moment when the sample was collected, whilst the no pneumonia panel group used the time of pneumonia diagnosis. To assess 30-day mortality in a valid and scientifically rigorous manner, the starting point for the calculation must be consistent across both groups (e.g. pneumonia diagnosis).

-     Differences between patients evaluated by the infectious disease team and those assessed by treating physicians are not described. Which are the referral criteria? Additionally, the authors do not specify whether the four centers follow the same protocols for pneumonia diagnosis and treatment, nor the criteria these centers use to decide whether to employ the pneumonia panel or refer patients to the infectious disease team. Is it part of routine clinical practice? In which specific cases is it used?

-     The authors should consider adding a microbiology subsection to describe the diagnostic techniques employed.

-     Tracheal aspirate, bronchoalveolar lavage, and sputum samples were included. The diagnostic performance (sensitivity and specificity) of the diagnostic method varies with sample type and should be taken into account in the analyses.

-     For assessing agreement between the pneumonia panel and standard culture, the authors used samples taken either simultaneously or with a maximum difference of 72 hours. The same sample should be used to evaluate agreement between diagnostic techniques to avoid bias (lines 263–266).

-     There is a lack of adjustment for heterogeneity between pneumonia types. In this context, balancing by IPTW may be insufficient. Since the sample size may be limited for stratified analysis by pneumonia type, the authors should consider including pneumonia type as a covariate in their models or focusing solely on data from severe CAP patients.

Finally, the results and conclusions sections lack clarity and present some inconsistencies in drawn that should be resolved to enhance the overall quality of the manuscript. Additionally, these sections should be aligned with the aims, as the conclusions drawn seem to highlight the importance of infectious disease teams rather than focusing on analyzing the impact of the pneumonia panel application, which differs from the stated aims.

Attention to these details will significantly improve the quality, consistency and clarity of the work.

Specific comments

The conclusion of the abstract (lines 30-33) presents two contradictory statements.

Non-standard abbreviations have been detected, such as those for health care-associated pneumonia (line 237) or hospital-acquired pneumonia (line 247). The use of standard abbreviations should be considered to ensure clarity and alignment with current literature. Furthermore, in this context, the term "exposure" (line 246) to refer to the use or non-use of the pneumonia panel may confuse the reader.

How is “appropriate antibiotic” defined in both groups (especially in the pneumonia panel group)?

Was the quality of the respiratory samples evaluated (especially those assessed by the pneumonia panel)?

What statistical program(s) did the authors use? Was any sensitivity analysis performed to test the robustness of the results with the IPTW?

Tables and figures lack comprehensive legends, which hinders the reader's ability to fully understand and interpret the data presented.

Figure 1 refers to 30-day mortality, but the graph describes 30-day mortality (days). The authors should verify whether this is a terminology error, an issue with the graphs, or a discrepancy in the data employed.

The text presents an unclear order when referencing the tables (lines 114-127). Also, lines 120-121 refer to Table 3, but the results mentioned (binomial model) do not appear in this Table. In fact, in Table 3 there are significant differences between the groups regarding the length of antibiotic therapy .

Regarding Table 3, are there more antibiotic changes in the no pneumonia panel group?

Why Table 4 does not show p-values and the same variables in the 5 main outcomes? What are the reference levels of these variables?

The use of performance in this context (line 150) may confuse the reader as it usually refers to the sensitivity, specificity, positive predictive value, negative predictive value, accuracy, and/or AUC of the test.

Lines 164-171 describe concepts that seem contradictory.

Author Response

Comment 1: The authors provide an introduction that includes current references, but pneumonia or severe pneumonia clinical guidelines are not mentioned. Additionally, the use of abbreviations should be reviewed, as some are not defined in the text and they use some non-standard abbreviations, which may hinder the manuscript understanding.

Answer 1: Thanks for the commentary. The guidelines, both the American and more recent European guidelines, were included (lines 50-54). Abbreviations were reviewed and, standard abbreviation were used when available.

Comment 2: There are significant concerns with the methodology employed that must be carefully addressed:

The authors used different timepoints to calculate the 30-day mortality (which is one of the primary outcomes); the pneumonia panel group used the moment when the sample was collected, whilst the no pneumonia panel group used the time of pneumonia diagnosis. To assess 30-day mortality in a valid and scientifically rigorous manner, the starting point for the calculation must be consistent across both groups (e.g. pneumonia diagnosis).

Answer 2: Thanks for calling our attention to this bias. The database was run again with the time of the diagnosis as the time to calculate the mortality and the hospital and ICU stay. The 30-day mortality did not change (70 cases in patients without the PN panel and 77 in those with the PN panel). The reason is that,although there were some cases with differences in the date of sampling (in relation to the diagnosis date), the mean of that difference was 0.2 days and the median was 1 day, not affecting the final outcome for mortality. The methods section was also changed accordingly (lines 356-358)

Comment 3: Differences between patients evaluated by the infectious disease team and those assessed by treating physicians are not described. Which are the referral criteria? Additionally, the authors do not specify whether the four centers follow the same protocols for pneumonia diagnosis and treatment, nor the criteria these centers use to decide whether to employ the pneumonia panel or refer patients to the infectious disease team. Is it part of routine clinical practice? In which specific cases is it used?

Answer: Thanks for this commentary. We added the variables that were different between those evaluated or not by the infectious diseases team (lines 112-117). The four centers have similar protocols for diagnosis and treatment of pneumonia and covid (lines 300-301).

Comment 4:The authors should consider adding a microbiology subsection to describe the diagnostic techniques employed.

Answer 4: Thanks for this observation. The information about the microbiological procedures was added (lines 276-293)

Comment 5:Tracheal aspirate, bronchoalveolar lavage, and sputum samples were included. The diagnostic performance (sensitivity and specificity) of the diagnostic method varies with sample type and should be taken into account in the analyses.

Answer 5: Thanks for this observation. The diagnostic performance of the diagnostic methods was included a priori. More information fo the culture breakpoints for reporting was added (lines 285-293)

Comment 6: For assessing agreement between the pneumonia panel and standard culture, the authors used samples taken either simultaneously or with a maximum difference of 72 hours. The same sample should be used to evaluate agreement between diagnostic techniques to avoid bias (lines 263–266).

Answer 6: Thanks for this comment, with which we agree with the evaluator. However, we recognize that it does not occur in real life in the same way. That is why we have chosen a limit of 72 hours of difference between the two samples.

Comment 7:There is a lack of adjustment for heterogeneity between pneumonia types. In this context, balancing by IPTW may be insufficient. Since the sample size may be limited for stratified analysis by pneumonia type, the authors should consider including pneumonia type as a covariate in their models or focusing solely on data from severe CAP patients.

Answer 7: The evaluator is right, this is an important variable with a potential effect on the outcomes. We included the variable in the model used to obtain the propensity. Additionally, we performed a new sensitivity analysis on the main outcome (30-day mortality), selecting only patients with CAP. The procedure was added to the methods (lines 356-358) and the results were also added (lines 141-144).

Comment 8:Finally, the results and conclusions sections lack clarity and present some inconsistencies in drawn that should be resolved to enhance the overall quality of the manuscript. Additionally, these sections should be aligned with the aims, as the conclusions drawn seem to highlight the importance of infectious disease teams rather than focusing on analyzing the impact of the pneumonia panel application, which differs from the stated aims.

Answer 8: Thanks a lot for your suggestion.  The section was changed according to the aims. Besides, the introduction emphasized a secondary outcome of the impact of Infectious Diseases Team interventions  Lines (68-70).

Comment 9:The abstract's conclusion (lines 30-33) presents two contradictory statements.

Answer 9: Thanks for this observation. We changed the abstract and conclusions to reflect the results according to the study's aims. Lines 30-34

Comment 10:Non-standard abbreviations, such as those for health care-associated pneumonia (line 237) or hospital-acquired pneumonia (line 247), have been detected. The use of standard abbreviations should be

considered to ensure clarity and alignment with current literature. Furthermore, in this context, the term "exposure" (line 246) to refer to the use or non-use of the pneumonia panel may confuse the reader.

Answer 10: The evaluator is right. According to current recommendations, the definition was changed to hospital-acquired pneumonia and ventilator-associated pneumonia (Kalil et al., Clinical Infectious Diseases  2016;63(5):e61–111) (line 86). We also changed the "exposure" term for "PN-panel" for simplicity and to avoid confusion.

Comment 11:How is “appropriate antibiotic” defined in both groups (especially in the pneumonia panel group)?

Answer 11: An appropriate antibiotic was defined as one that was susceptible in vitro when susceptibility testing was available, the absence of a mechanism of resistance, or the recommended treatment according to the mechanism of resistance identified by the molecular panel. This was added to the methodology section (lines 333-337).

Comment 12:Was the quality of the respiratory samples evaluated (especially those assessed by the pneumonia panel)?

Answer 12: Thanks for this observation. As mentioned before, pre-defined quality assessment and breakpoints for the report were established  according to standardized protocol in the  microbiology labs (lines 276-293)

Comment 13:What statistical program(s) did the authors use? Was any sensitivity analysis performed to test the robustness of the results with the IPTW?

Answer 13: Thanks for the recommendation. The information was added in lines 356-358. No sensitivity analysis was performed in the IPTW results since the recommended steps for evaluating the statistical procedure's results were followed, including evaluating the balance of the groups (graphically and with the SMD).

Comment 14:Tables and figures lack comprehensive legends, which hinders the reader's ability to fully understand and interpret the data presented.

Answer 14: Legends were added to the tables fo facilitate the interpretations.

Comment 15:Figure 1 refers to 30-day mortality, but the graph describes 30-day mortality (days). The authors should verify whether this is a terminology error, an issue with the graphs, or a discrepancy in the data employed.

Answer 15: Thank you for calling our attention to this mistake, the graph was corrected.

Comment 16:The text presents an unclear order when referencing the tables (lines 114-127). Also, lines 120-121 refer to Table 3, but the results mentioned (binomial model) do not appear in this Table.

Answer 16: The evaluator is right. The mistake was corrected (lines 146 & 152) and the correct table referenced.

Comment 17:In fact, in Table 3 there are significant differences between the groups regarding the length of antibiotic therapy . Regarding Table 3, are there more antibiotic changes in the no pneumonia panel group?

Answer 17: Yes, we observed more changes in the no PN-panel group.

Comment 18:Why Table 4 does not show p-values and the same variables in the 5 main outcomes? What are the reference levels of these variables?

Answer 18: p-values were added to the table. For simplicity of the models, when other variables different from the study variable (PN-panel use) did not reach statistical significance, they were not included in the model. We anotated the reference level used for each model for each variable.

Comment 19:The use of performance in this context (line 150) may confuse the reader as it usually refers to the sensitivity, specificity, positive predictive value, negative predictive value, accuracy, and/or AUC of

the test.

Answer 19: The term performace was changed for "use" (line 176)

Comment 20:Lines 164-171 describe concepts that seem contradictory.

Answer 20:Thanks  a lot for your comments. The paragraph was modified (lines 186-196)

Reviewer 2 Report

Comments and Suggestions for Authors

The submitted manuscript, which represents an interesting contribution to the treatment of critically ill patients with pneumonia, fits into the focus of Antibiotics. It describes the significance of the multiplex PCR pneumonia panel (PN-panel) in patients with pneumonia on mortality and other important clinical outcomes.

Unfortunately, in its current state, I cannot recommend the manuscript for acceptance and, in my opinion, major text modifications are necessary.

Major comments

It is necessary to supplement the clinical and laboratory criteria for establishing the diagnosis of community-acquired, hospital-acquired and ventilation-associated pneumonia. How were these diagnoses confirmed?

I consider it appropriate to supplement the PCT values.

How was the definition of "Appropriate antibiotic" determined?

It is appropriate to supplement the definitions of the terms "Empiric antibiotics" and "Narrow spectrum antibiotics".

A precise description of the microbiological diagnostics used is missing. A precise description of the PN-panel must be supplemented. How were blood cultures examined and how were clinically insignificant findings excluded. The stated etiology of Staphylococcus epidermidis is very unlikely. Were the results of the PN-panel confirmed by standard microbiological examination? The same applies to the determination of bacterial susceptibility/resistance to antibiotics, were phenotypic methods also used?

Minor comments

Covid-19 is an infection and not an etiological agent. The sentence "COVID-19 was the primary etiologic agent" is not correct. The etiological agent is SARS-CoV-2.

Bacterial names are always written in italics.

The term "staphylococci" is written with a lowercase "s". It is not a genus name.

What does the abbreviation MRJE mean?

Author Response

Comment 1:It is necessary to supplement the clinical and laboratory criteria for establishing the diagnosis of community-acquired, hospital-acquired and ventilation-associated pneumonia. How were these diagnoses confirmed?

Answer 1: Centers and physicians followed local guidelines (Montúfar et al Infectio 2013; 17 (supl 1): 1-38. and Cortés JA et al. Rev Fac Med 2022; 70 (2):e93814. https://doi.org/10.15446/revfacmed.v70n2.93814 and Saavedra et al Infectio 2020; 22(3 supl): 1-102 for COVID) that were adapatations from the American guidelines(American Journal of Respiratory and Critical Care Medicine, Volume 200, Issue 7, 1 October 2019). In short, for a diagnosis of pneumonia, the patient required the presence of respiratory symptoms, clinical signs of pulmonary compromise, a chest X-ray or other pulmonary image with a consolidation pattern. This was added to the methods: lines 263--265.

Comment 2:I consider it appropriate to supplement the PCT values.

Answer 2: Thanks for the recommendation, that we think is very valuable. However, this information was not collected.

Comment 3:How was the definition of "Appropriate antibiotic" determined?

Answer 3: Appropriate antibiotic was defined as an antibiotic susceptible in vitro, when the susceptibility testing was available,  the abscence of a mechanism of resistance or the recommended treatment according to the mechanism of resistance identified by the molecular panel. This was added to the methodology section (lines 333-337).

Comment 4:It is appropriate to supplement the definitions of the terms "Empiric antibiotics" and "Narrow spectrum antibiotics".

Answer 4: Thanks for the observation. Defintions on "empiric antibiotics" and "narrow spectrum" were added (based on Warrell D et al. Oxford Textbook of Medicine: Infection (2012). OUP Oxford. p. 39). (lines 338-341)

Comment 5:A precise description of the microbiological diagnostics used is missing. A precise description of the PN-panel must be supplemented.

Answer 5: Information on microbiological procedures, specially the PN panel was added (lines 276-293)

Comment 6:How were blood cultures examined and how were clinically insignificant findings excluded. The stated etiology of Staphylococcus epidermidis is very unlikely.

Answer 6: Thanks for the observations, which is correct. It was stated that this result was interpreted as contamination (line 102).

Comment 7:Were the results of the PN-panel confirmed by standard microbiological examination? The same applies to the determination of bacterial susceptibility/resistance to antibiotics, were phenotypic methods also used?

Answer 7: Microbiological procedures were added to the methods section (lines 276-293).

Comment 8:Covid-19 is an infection and not an etiological agent. The sentence" COVID-19 was the primary etiologic agent" is not correct. The etiological agent is SARS-CoV-2.

Answer 8: The evaluator is right and the correction was done (line 25).

Comment 9:Bacterial names are always written in italics.

Answer 9: Yes, the corrections were done.

Comment 10:The term "staphylococci" is written with a lowercase "s". It is not a genus name.

Answer 10: According to the Merriam -Webster dictionary staphylococci is written with a lowercase "s". We found the same use in the National Library of Medicine.

Comment 11:What does the abbreviation MRJE mean?

Answer 11: The evaluator is right, the correct name is MREJ (staphylococcal cassette chromosome mec [SCCmec]-orfX right-extremity junction). The correction was done (line 106-107).

Round 2

Reviewer 1 Report

Comments and Suggestions for Authors

Brief summary of the manuscript

The authors aimed to investigate the effect of a multiplex PCR panel for pneumonia (Filmarray®, Biomérieux, France) on hospital stay, intensive care unit (ICU) stay, duration of antimicrobial treatment, and 30-day mortality in ICU patients with severe pneumonia through a retrospective cohort of 304 patients between 2019 and 2023. They did not find an association between the primary outcomes and the use of the pneumonia panel, and the conclusions drawn suggest the application of the pneumonia panel must be accompanied by the clinical judgement of an infectious diseases team.

General concept comments

The topic addressed is both interesting and relevant, reflecting an important area of research. Regarding the major comment on the agreement between the pneumonia panel and standard culture, it’s important to note that the same sample or samples taken in close temporal proximity should have been used to minimize bias. As this issue couldn’t be fully resolved as it is a retrospective cohort, it should be mentioned in the limitations section, as the use of different samples within a 72-hour window may introduce bias. This timeframe is quite broad, especially considering that patients are receiving antibiotics.

Specific comments

  1. Lines 32-34: “However”, “nevertheless” I still feel like there’s a missing link between these two statements. It would be helpful to add a sentence explaining how the second idea follows from the first to improve clarity.
  2. Line 60: PCR has not been previously defined.
  3. Line 76: The “.0” can be omitted as it doesn’t add extra information.
  4. Lines 121-124: In the text, the descriptive analysis lacks numerical data. While these values are included in Table 1, explicitly mentioning them would help the reader better interpret statements (mean or median with Q1-Q3, and p-values, or n(%) with p-values).
  5. Line 152: What’s the meaning of *?
  6. Line 224: Point of care has not been previously defined.
  7. Line 264: CRP or PCR?
  8. Lines 392-393: R is not a program but a programming language (the authors can simply say “using R” instead).
  9. The issue regarding the use of the term 'exposure' (line 246) to refer to the use or non-use of the pneumonia panel has been addressed. However, I still see it being used in some parts of the text. Could you double-check for consistency? For example, in lines 76, 109, 160-167.
  10. Please review the use of “vs.”; in several cases, it appears as 'vs' instead of 'vs.'. For example, in lines 127-129.
  11. Please review the use of the abbreviation 'ID' for Infectious Disease. In several occasions within the text (excluding titles), the full term is used instead of the abbreviation, and its definition appears quite late (first mentioned in line 72). It would be clearer to define it earlier and ensure consistent use throughout the text. For example, in lines 147, 331.
  12. Please review the use of dashes for numerical ranges, as I noticed some cases where a double hyphen ('--') is used (e.g. lines 158, 176).
  13. In general, I still find the table captions quite minimal. Adding a bit more detail could make them clearer and more informative.
Comments on the Quality of English Language
  1. Line 25: SARS-CoV-2.
  2. Table 1 and lines 94-98: Please review the use of capitalization.
  3. Lines 118-124: Please review this section, as there are several typos.
  4. Line 139: are shown.
  5. Line 393:
  6. Please review the text, as I noticed some instances where 'COVID' is used instead of 'COVID-19'. 'COVID-19' is more appropriate.

Author Response

General concept comments

Comment 1: The topic addressed is both interesting and relevant, reflecting an important area of research. Regarding the major comment on the agreement between the pneumonia panel and standard culture, it’s important to note that the same sample or samples taken in close temporal proximity should have been used to minimize bias. As this issue couldn’t be fully resolved as it is a retrospective cohort, it should be mentioned in the limitations section, as the use of different samples within a 72-hour window may introduce bias. This timeframe is quite broad, especially considering that patients are receiving antibiotics.

Answer 1: Thanks for the comment. This limitation was pointed in the limitations segment of the discussion, lines 245-249.

Specific comments:

Comment 2: Lines 32-34: “However”, “nevertheless”… I still feel like there’s a missing link between these two statements. It would be helpful to add a sentence explaining how the second idea follows from the first to improve clarity.

Answer 2: The phrase was changed to improve clarity.

Comment 2: Line 60: PCR has not been previously defined.

Answer 3: The defintion was provided.

Comment 3: Line 76: The “.0” can be omitted as it doesn’t add extra information.

Answer 4: The change was made.

Comment 4: Lines 121-124: In the text, the descriptive analysis lacks numerical data. While these values are included in Table 1, explicitly mentioning them would help the reader better interpret statements (mean or median with Q1-Q3, and p-values, or n(%) with p-values).

Answer 4: Information was added.

Comment 5: Line 152: What’s the meaning of *?

Answer 5: Thanks for call our attention to it. It has no meaning and was retired.

Comment 6: Line 224: Point of care has not been previously defined.

Answer 6: The definition was made and the abbreviation was retired.

Comment 7: Line 264: CRP or PCR?

Answer 7: PCR, the change wa made.

Comment 8:Lines 392-393: R is not a program but a programming language (the authors can simply say “using R” instead).

Answer 8: The correction was made.

Comment 9:The issue regarding the use of the term 'exposure' (line 246) to refer to the use or non-use of the pneumonia panel has been addressed. However, I still see it being used in some parts of the text. Could you double-check for consistency? For example, in lines 76, 109, 160-167.

Answer 9: We double-checked, an the term is not used any more in that sense.

Comment 10:Please review the use of “vs.”; in several cases, it appears as 'vs' instead of 'vs.'. For example, in lines 127-129.

Answer 10: Thanks for calling our attention ot it, it was changed accordingly.

Comment 11: Please review the use of the abbreviation 'ID' for Infectious Disease. In several occasions within the text (excluding titles), the full term is used instead of the abbreviation, and its definition appears quite late (first mentioned in line 72). It would be clearer to define it earlier and ensure consistent use throughout the text. For example, in lines 147, 331.

Answer 11: We changed the use of the abbreviation.

Comment 12: Please review the use of dashes for numerical ranges, as I noticed some cases where a double hyphen ('--') is used (e.g. lines 158, 176).

Answer 12: It was corrected.

Comment 13: In general, I still find the table captions quite minimal. Adding a bit more detail could make them clearer and more informative.

Answer 13: Information was added to tables 2,3, and 4

Comments on the Quality of English Language

Comment 14: Line 25: SARS-CoV-2.

Answer 14: The change was done.

Comment 15: Table 1 and lines 94-98: Please review the use of capitalization.

Answer 15: Capitalization was reviewed in figure 1, table 1 and the lines indicated.

Comment 16: Lines 118-124: Please review this section, as there are several typos

Answer 16: The section was reviewed.

Comment 17:Line 139: are shown.

Answer 17: Correction was done.

Comment 18: Line 393: Please review the text, as I noticed some instances where 'COVID' is used instead of 'COVID-19'. 'COVID-19' is more appropriate.

Answer 18: The change was made.

Reviewer 2 Report

Comments and Suggestions for Authors

First of all, I thank the authors of the manuscript for editing the text based on my comments.

The text has been adequately edited and I am now pleased to recommend the manuscript for acceptance.

Author Response

Thanks.